# A Retrospective Study of *Staphylococcus aureus* Bacteremia in a Tertiary Hospital and Factors Associated with Mortality

**DOI:** 10.3390/diagnostics13111975

**Published:** 2023-06-05

**Authors:** Petros Ioannou, Maria Zacharioudaki, Despoina Spentzouri, Antonia Koutoulakou, Konstantinos Kitsos-Kalyvianakis, Christoforos Chontos, Stamatis Karakonstantis, Sofia Maraki, George Samonis, Diamantis P. Kofteridis

**Affiliations:** 1School of Medicine, University of Crete, 71003 Heraklion, Greece; 2Internal Medicine Department, University Hospital of Heraklion, 71110 Heraklion, Greece; 3Pediatrics Department, University Hospital of Heraklion, 71110 Heraklion, Greece; 4Department of Clinical Microbiology, University Hospital of Heraklion, 71110 Heraklion, Greece

**Keywords:** *Staphylococcus aureus*, bacteremia, bloodstream infection, antimicrobial resistance, mortality

## Abstract

*Staphylococcus aureus* bacteremia (SAB) is a severe infection frequently associated with significant morbidity and mortality. Recent studies have shown that SAB mortality has decreased during the last decades. However, about 25% of patients suffering from the disease will ultimately die. Hence, there is an urgent need for more timely and efficient treatment of patients with SAB. The aim of the present study was to retrospectively evaluate a cohort of SAB patients hospitalized in a tertiary hospital and to identify factors independently associated with mortality. All 256 SAB patients hospitalized from January 2005 to December 2021 in the University Hospital of Heraklion, Greece, were evaluated. Their median age was 72 years, while 101 (39.5%) were female. Most SAB patients were cared for in medical wards (80.5%). The infection was community-acquired in 49.5%. Among all strains 37.9% were methicillin-resistant *S. aureus* (MRSA), however, definite treatment with an antistaphylococcal penicillin was given only in 22% of patients. Only 14.4% of patients had a repeat blood culture after the initiation of antimicrobial treatment. Infective endocarditis was present in 8%. In-hospital mortality has reached 15.9%. Female gender, older age, higher McCabe score, previous antimicrobial use, presence of a central venous catheter, neutropenia, severe sepsis, septic shock, and MRSA SAB were positively associated with in-hospital mortality, while monomicrobial bacteremia was negatively associated. The multivariate logistic regression model identified only severe sepsis (*p* = 0.05, odds ratio = 12.294) and septic shock (*p* = 0.007, odds ratio 57.18) to be independently positively associated with in-hospital mortality. The evaluation revealed high rates of inappropriate empirical antimicrobial treatment and non-adherence to guidelines, as shown, by the lack of repeat blood cultures. These data underline the urgent need for interventions with antimicrobial stewardship, increased involvement of infectious diseases physicians, educational sessions, and creation and implementation of local guidelines for improvement of the necessary steps for timely and efficient SAB treatment. Optimization of diagnostic techniques is needed to overcome challenges such as heteroresistance that may affect treatment. Clinicians should be aware of the factors associated with mortality in patients with SAB to identify those who are at a higher risk and optimize medical management.

## 1. Introduction

*Staphylococcus aureus* bacteremia (SAB) is an important and relatively frequent infection associated with significant morbidity and mortality [1]. The annual incidence depends on the country where data are collected. Hence, in a study from Canada in 2008, an annual incidence of 19.7 cases per 100,000 people was estimated, while, in Scandinavian countries, this incidence is approximately 26 per 100,000 population [2,3,4]. However, in other countries where the incidence of methicillin-resistant *S. aureus* (MRSA) is higher, the rates of SAB are reportedly higher, estimated at 35–39 per 100,000 population, but may be as high as 50 per 100,000 population, as shown in a study from the USA [5,6,7,8]. This discrepancy may not be associated only with the incidence of methicillin-resistant *Staphylococcus aureus* (MRSA), but may also be associated with other factors, such as infection control practices, adequacy of data collection, and differences in healthcare systems [1].

SAB incidence is higher as age increases, being lower in pediatric populations and among young adults with an incidence of 10 per 100,000 [2,4,9]. Male gender, intravenous drug use, infection by the human immunodeficiency virus (HIV), nasal colonization, other host factors, such as ethnicity and past medical history (e.g., history of diabetes mellitus), hospitalization, and frequent contact with the healthcare system, such as in patients with renal replacement therapy are also factors associated with a higher likelihood for SAB development [2,7,10,11,12,13,14,15,16].

SAB is associated with high mortality, especially in critically ill patients hospitalized in the intensive care unit (ICU) [1,17,18]. Before the antibiotic era, this infection had a mortality that could be close to 80% [19]. With the development of current antimicrobial treatment mortality rates are becoming quite lower, reaching 20% for 30-day mortality, while according to recent studies, infection-related mortality reaches 13% [20]. A recent systematic review showed that SAB mortality may have decreased in the last decades, however, more than 25% of patients will die within three months, thus, there is still a need for further improvement [21]. Improvement of mortality rates would require a better understanding of its risk factors, as well as optimization of issues regarding appropriate treatment as well as diagnosis, especially in terms of microbiology. For example, heteroresistance, which is the resistance to specific antimicrobials expressed by only a subset of a microbial population that is otherwise considered to be susceptible to these drugs according to the classic in vitro susceptibility testing is a problem unresolved based on the current routine microbiological techniques [22]. In terms of risk factors, mortality differed among different patients, with patients with underlying comorbidities, MRSA infection, and time of blood culture positivity of less than 12 h having higher mortality [1,20,21,23,24].

For further improvement of these patients’ care, it is important to identify those being at higher risk for worst outcomes through the identification of factors associated with mortality. Thus, the aim of the present study was to retrospectively evaluate a cohort of patients with SAB hospitalized in a tertiary hospital and identify factors independently associated with mortality.

## 2. Materials and Methods

### 2.1. Study Type and Ethics Approval

This is a retrospective single-center cohort study of patients with SAB who were hospitalized from January 2005 to December 2021 in the University Hospital of Heraklion, Heraklion, Crete, Greece, a tertiary hospital with 771 beds. Patients were included in the study if they had bacteremia by *S. aureus*, which was defined as the presence of at least one positive blood culture for *S. aureus* during their hospitalization. There was no exclusion criterion. The primary outcome of the present study was to provide data regarding mortality and identify the factors that are independently associated with it. Secondary outcomes included the description of epidemiology, microbiology, and treatment of SAB, and identification of any differences in these parameters before and after the coronavirus disease-2019 (COVID-19) era. Data collected and evaluated included patients’ age, gender, medical history, McCabe score (a score used in epidemiological studies that stratifies patients depending on whether they suffer from a non-fatal (1 point), a rapidly fatal (2 points), or an ultimately fatal disease (3 points)) [25], duration of hospitalization, the ward where the blood culture was drawn, the outcome of hospitalization, antimicrobial resistance of *S. aureus*, type of blood culture (monomicrobial or not), other isolated microorganisms, whether the infection was primary bloodstream infection (BSI), co-existing skin and soft tissue infection (SSTI), endocarditis or other, and whether the infection was community-acquired or hospital-acquired. The data regarding patients with bacteremia were provided by the microbiology department and the rest of the data were retrieved from the hard copies of the patients’ notes and the hospital’s electronic system. Infection was considered community-acquired if the blood culture was drawn up to 48 h after admission. Post-COVID-19 era was defined as the era from 2020 until the end of the study. Empirical treatment was deemed appropriate if it included at least one antimicrobial agent active against the *S. aureus* strain that was eventually identified. The study follows the guidelines for reporting observational studies (Strengthening the Reporting of Observational Studies in Epidemiology—STROBE)—Appendix A [26].

The study has been approved by the Institutional Review Board of the University Hospital of Heraklion.

### 2.2. Sample Collection, Transport, and Processing

Blood was collected in blood culture bottles that were promptly transported to the microbiology laboratory for further processing. Bottles were loaded and incubated on the BacT/Alert Virtuo system (BioMérieux, Marcy L’Étoile, France) for five days unless growth was detected earlier. When a culture bottle had been signaled positive, gram stain and subcultures were immediately performed. For the isolation of bacterial pathogens, specimens were inoculated onto Columbia blood, chocolate, MaC Conkey, and Schaedler blood agar (all products of bioMérieux SA, Marcy L’Étoile, France) and incubated at 36 °C. Identification of bacterial species was performed by standard biochemical assays and the Vitek 2 automated system and confirmed by the matrix-assisted laser desorption time of flight, mass spectrometry (MALDI-TOF MS) (version 3.2) (both products of bioMérieux SA). The Vitek 2 automated system was also used for antimicrobial susceptibility testing and results were interpreted according to the Clinical and Laboratory Standards Institute (CLSI) criteria [27].

### 2.3. Statistics

Descriptive statistics were performed with GraphPad Prism 6.0 (GraphPad Software, Inc., San Diego, CA, USA). Categorical data were analyzed with Fisher’s exact test. Continuous variables were compared using the Mann–Whitney U-test for non-normally distributed variables. All tests were two-tailed and a *p*-value equal to or lower than 0.05 was considered significant. Data are presented as numbers (%) for categorical variables and medians (interquartile range (IQR)) for continuous variables. A linear regression analysis model was developed to evaluate the effect of several parameters [age, gender, McCabe score, previous hospitalization, previous surgery, previous antimicrobial use, presence of a central venous catheter (CVC), community-acquired state, renal replacement therapy (RRT), neutropenia, presence of at least two severe inflammatory response syndrome (SIRS) criteria [28], severe sepsis or septic shock, infective endocarditis, monomicrobial BSI, MRSA, and duration of treatment] with in-hospital mortality. All parameters were calculated with GraphPad Prism 6.0 (GraphPad Software, Inc., San Diego, CA, USA). A multivariate logistic regression analysis model was developed to evaluate the association of factors identified in the univariate analysis with a *p* lower than or equal to 0.1 with mortality. Multivariate analysis was performed using the SPSS version 23.0 (IBM Corp., Armonk, NY, USA), and a *p*-value equal to or lower than 0.05 was considered significant (along with a confidence interval of 95%).

## 3. Results

### 3.1. Patients’ Characteristics

In total, 256 patients had an episode of SAB during their hospitalization at the University Hospital of Heraklion during the study period. No patients were excluded from the analysis. Patients had a median age of 72 years and 101 (39.5%) were female. Their medical conditions can be seen in Appendix A. The positive blood culture was most commonly drawn in a medical ward (80.5%), followed by a surgical ward (13.5%) and the ICU (6%). SAB was community-acquired in 49.5%. Among all *S. aureus* strains 37.9% were MRSA. A repeat blood culture was taken only in 14.4% of patients after initiation of antimicrobial treatment and was sterile in 84.6%. The median duration of stay in the hospital was 20 days, and in-hospital mortality was 15.9%.

Patients who died were of older age, were more likely to be female, had a higher McCabe score, were less likely to be hospitalized in a surgical ward and more likely to be hospitalized in the ICU, and to have a CVC, total parenteral nutrition (TPN), neutropenia, severe sepsis, and septic shock. Furthermore, patients who died were also more likely to have polymicrobial bacteremia and to have bacteremia by MRSA. Teicoplanin was more commonly used among patients who died.

Table 1 shows the characteristics of patients with SAB in total and in regards to whether they survived or died, while Table 2 shows the treatment and outcomes of patients with SAB in total and in regards to whether they survived or died.

A comparison of patients with SAB before and after the onset of the COVID-19 pandemic revealed that patients had similar characteristics with the exception that infective endocarditis was more common in patients with SAB in the post-COVID-19 era, while central-line associated bloodstream infections (CLABSIs) were more common in the pre-COVID-19 era. In terms of treatment and outcomes, the duration of antimicrobial use and hospitalization, was longer in patients with SAB in the post-COVID-19 era, while teicoplanin was not used in the post-COVID-19 era. Mortality was similar in these two patient populations. Table 3 andTable 4 show the characteristics, treatment, and outcomes of patients with SAB in regard to when the bacteremia occurred.

### 3.2. Regression Analysis of In-Hospital Mortality among Patients with SAB

To identify factors associated with in-hospital mortality, a regression analysis was performed. First, a univariate linear regression analysis was performed to evaluate the effect of several parameters such as age, gender, McCabe score, previous hospitalization, previous surgery, previous anti-microbial use, presence of a CVC, presence of TPN, RRT, neutropenia, presence of at least two SIRS criteria, severe sepsis or septic shock, infective endocarditis, monomicrobial bacteremia, MRSA, duration of treatment, and duration of hospital stay with in-hospital mortality. Female gender, higher age, higher McCabe score, previous antimicrobial use, presence of a CVC, neutropenia, severe sepsis, septic shock, and SAB by MRSA were positively associated, while monomicrobial bacteremia was negatively associated with in-hospital mortality. However, a multivariate logistic-regression model identified only severe sepsis (*p* = 0.05, odds ratio = 12.294) and septic shock (*p* = 0.007, odds ratio 57.18) to be independently positively associated with in-hospital mortality. Table 5 shows the results of the regression analysis of in-hospital mortality among patients with SAB.

## 4. Discussion

This study presented data from a retrospective cohort of patients with SAB hospitalized at the University Hospital of Heraklion. In total, 256 patients with SAB were recorded and evaluated. SAB was community-acquired in about half the cases and about 38% of the strains were MRSA. Infective endocarditis was diagnosed only in 8% of patients with SAB. Hospital mortality was 16%, while, among the several factors associated with in-hospital mortality, the development of severe sepsis or septic shock was independently associated with this outcome. Patients with SAB before and after the onset of the COVID-19 pandemic had similar characteristics with the exception that infective endocarditis was more commonly diagnosed in the post-COVID-19 era, while the duration of antimicrobial use and hospitalization, was also longer in patients of this time period.

SAB is a severe infection and cause of significant morbidity and mortality. The 30-day all-cause mortality is approximately 15–20% [1,29,30,31,32]. In the present study, in-hospital mortality was 16%, in accordance with that reported by other investigators, even though, in the present study, 30-day mortality could not be evaluated [31,33]. In the present study, a regression analysis identified several parameters associated with in-hospital mortality. More specifically, these parameters were female gender, age, McCabe score, previous antimicrobial use, presence of CVC, neutropenia, severe sepsis, septic shock, and polymicrobial BSI. The McCabe score is a tool that may be of use for its prognostic value and is used to predict mortality in hospitalized patients based on their underlying medical conditions. In patients with bacteremia, it can be useful in predicting the outcome of infection. In another study evaluating outcomes of patients with BSI by MRSA, McCabe score was again found to differ significantly among patients who died and those who survived. In particular, patients who died had a worse McCabe score, and this score was also identified in a regression analysis to be independently associated with mortality [34]. In the same study, older age was also associated with higher mortality, as well as previous hospitalization, as in the present study in a statistically significant way [34]. Importantly, in that study, delay of appropriate antimicrobial treatment was the most significant factor associated with mortality. Paradoxically, in the present study, the rate of appropriate empirical treatment was not significantly different among patients who survived and those who died. Only severe sepsis and septic shock in the present patients were independently associated with mortality in the multivariate analysis. In line with our findings, other studies have identified septic shock as a determinant of 90-day infection-related mortality, along with age, Charlson comorbidity index, endocarditis, and persistent bacteremia at 48 h [31,35]. In other studies, mortality was found to be higher in patients who had underlying comorbidities, in those with MRSA BSI, and in those where time to positivity of blood cultures was less than 12 h, while infectious diseases consultation has been associated with a lower likelihood for mortality [1,20,21,23,24,31,32].

SAB can be classified into three categories, healthcare-associated with hospital-onset (nosocomial), healthcare-associated with community-onset, and community-acquired. In a prospective cohort study with 247 cases in the USA, 23% of patients had SAB of nosocomial origin, 59% healthcare associated with community-onset, and 18% community-acquired SAB. In the present study, patients were classified as those who had nosocomial (hospital-acquired) and community-acquired SAB, with half the patients being classified in the first category. According to the literature, patients with community-acquired SAB are at a higher risk of complications, including the possibility of infective endocarditis. Hence, as reported by a similar study, more than 40% of patients with a community-acquired SAB had a metastatic infection, such as infective endocarditis [36]. The corresponding rate of metastatic complications for nosocomial SAB had been reported as being lower, at the level of 20% [1,36]. The rate of metastatic complications among the present patients was low. More specifically, infective endocarditis was diagnosed only in 8%.

Management of SAB includes the exclusion of infective endocarditis, adequate source control, such as removal of an infected CVC, and appropriate antimicrobial treatment [31,37,38,39]. To reduce mortality, empirical antimicrobial treatment should be started as soon as possible taking into account the local antimicrobial resistance patterns, and more specifically, the likelihood of MRSA SAB. In the present study, MRSA was identified in about 40% of the strains. Thus, it is not surprising that empirical antimicrobial treatment included vancomycin or daptomycin in most of the patients, while, these antimicrobials were also very commonly used as definite treatment, with antistaphylococcal penicillin being used in a lower proportion. However, even though 60% of SAB were due to MSSA, treatment with antistaphylococcal penicillin was used in less than 25%. Even though data regarding penicillin allergy were not available in the present study, the fact that less than 50% of patients eligible to be treated with antistaphylococcal penicillin were treated with another antimicrobial implies that rates of adequate, evidence-based treatment were probably not very high in this patient cohort [38]. Since treatment of bacteremia by MSSA with antimicrobials other than antistaphylococcal penicillin is associated with worst outcomes, it is of utmost importance the present data to ignite appropriate educational and structural changes that could lead to rationalization of antimicrobial treatment in such cases [40,41,42,43,44,45,46,47]. This could be performed by increasing the number of infectious diseases consults in patients with SAB. Another issue that arises from the present observations has to do with the finding that empirical treatment was not adequate in about 40% of patients. Adequate empirical treatment is of high significance since it has been shown to negatively affect mortality in SAB patients in several studies [34]. However, in some studies, inadequate empirical antimicrobial treatment was found to only slightly increase mortality in similar populations [48,49]. Thus, it is important to note that there may be a clear inherent limitation of the treatment strategy that is based only on empirical data. Indeed, there seems to be a clear need for evidence-based laboratory data, such as specific minimum-inhibitory concentrations for antimicrobials tested, from multiple time points during the course of patients’ treatment [50].

Notably, an issue that may negatively affect the management of SAB, as well as other infections, has to do with the way the issue of antimicrobial resistance is addressed. More specifically, heteroresistance is an important issue that may lead to misinterpretation of the antimicrobial resistance in SAB [22]. For *S. aureus* in specific, low-level resistance to methicillin may be present in the majority of bacterial cells, while, a minority may have high-level resistance to methicillin [51]. Thus, it would be of high practical significance, in an attempt to address this issue of heteroresistance, which may negatively affect the adequate treatment of SAB, to consider the implementation of changes in practice, either in terms of reconsideration of the present methodology in antimicrobial susceptibility testing using the newer technology [52].

Management of SAB may, at times, be challenging, especially in cases of persistent SAB. Recent reviews have focused on this particular problem and provided insights into strategies that could be employed for its treatment [53,54]. Beyond the classic approaches of source control and adequate antimicrobial therapy, novel treatment approaches exist, such as bacteriophages, antimicrobial peptides, or treatment with nanoparticles [55,56,57,58].

Consultation by infectious diseases physicians constitutes a core element in the management of SAB patients and should be performed whenever possible [59,60,61,62,63,64]. It is of note that the value of telephone consultation has been proven inferior to bedside consultation in such cases [63]. In the present study, infectious diseases consultations were provided only upon request from the treating physician. There are studies suggesting that the implementation of unsolicited infectious diseases consultation in the management of SAB patients may lead to better management, while it may also become an important intervention in terms of antimicrobial stewardship [65,66]. Implementation of a bundle-based intervention could increase adherence to evidence-based recommendations and allow for more consistent management of this lethal infection. In the present study, a repeat blood culture was drawn in only 14% of patients, a rate very low that makes urgent the need for the implementation of guidelines for the management of SAB through the application of interventions from infectious diseases and antimicrobial stewardship [37,38].

This retrospective study included patients hospitalized mostly before the onset of the COVID-19 pandemic, but it did also include some patients after its onset. Interestingly, the differences between these two patient subpopulations were small, but they did include a lower rate of CLABSIs in the post-COVID-19 era in terms of patients’ characteristics. However, the duration of antimicrobial use and hospitalization was longer in patients with SAB in the post-COVID-19 era. This could probably reflect the higher rate of diagnosis of infective endocarditis in the post-COVID-19 era among patients with SAB since treatment of infective endocarditis necessitates more prolonged antimicrobial treatment compared to an uncomplicated SAB [37,38].

The current study provides insights regarding the factors associated with mortality in patients with SAB. Since clinical severity of infection, and more specifically, presentation with severe sepsis or septic shock are independently associated with mortality, the results of the present study imply that optimization of management of SAB patients with this clinical severity is needed to reduce mortality. Clinicians caring for patients with severe sepsis or septic shock should consider the possibility of SAB early and provide adequate empirical antimicrobial coverage for *S. aureus* as well, since delay of treatment for these patients may significantly increase mortality. Furthermore, prompt responses for patients with severe sepsis or septic shock due to SAB with adequate source control are required, since these are the patients who are most at risk for worse outcomes.

This study has some notable limitations. First of all, it is a single-center study; thus, the results should be read with caution, since they represent the characteristics and the antimicrobial resistance patterns of the area where the study was performed. Furthermore, due to the retrospective nature of the study, some data were missing. For example, data about mortality were not available for 17 patients, as mentioned in the tables, while, only in-hospital mortality could be evaluated. Thus, 30-day mortality could not be estimated in the present patient cohort. Furthermore, since the time period of the study is from 2005 to 2021, some techniques in microbiology regarding isolation and identification may have changed during the study period. Finally, some factors such as age, the severity of current illness (e.g., presence of sepsis), the severity of underlying disease, and antimicrobial resistance could be confounding variables in the logistic regression analysis, thus, affecting results.

## 5. Conclusions

The present retrospective cohort study includes data on patients hospitalized with SAB before and during the COVID-19 pandemic. SAB was community-acquired in about half of the cases and about 38% of strains were MRSA. In-hospital mortality reached 16%, while, the development of severe sepsis or septic shock was independently associated with this outcome as shown by a multivariate regression analysis. Patients with SAB before and after the onset of the COVID-19 pandemic had similar characteristics with the exception that infective endocarditis was more commonly diagnosed in the post-COVID-19 era, with the duration of antimicrobial treatment and hospitalization being also longer in this patient population. Several remarks can be made regarding high rates of inappropriate empirical antimicrobial treatment and non-adherence to guidelines, as shown by the lack of repeat blood cultures. These data underline the urgent need for antimicrobial stewardship educational activities, increased involvement of infectious diseases physicians, even with unsolicited consultations, and creation and implementation of local guidelines through bundles to increase the uniform application of the necessary steps for timely and efficient treatment of SAB. Optimization of diagnostic techniques is needed to overcome challenges such as heteroresistance that may affect treatment. Finally, since clinical presentation with severe sepsis or septic shock is independently associated with mortality in patients with SAB, clinicians caring for these patients should optimize medical treatment by providing appropriate antimicrobial therapy, and also perform source control when applicable, to reduce mortality.

## Figures and Tables

**Table 1 diagnostics-13-01975-t001:** Characteristics of patients with *Staphylococcus aureus* bacteremia in total and in regards to mortality.

Characteristic	All Patients * (*n* = 256)	Survived (*n* = 201)	Died (*n* = 38)	*p*-Value ***
Age, years, median (IQR)	72 (60–82)	69 (54.5–80)	80 (68.5–85.3)	0.0005
Female gender, *n* (%)	101 (39.5) **	75 (37.3)	21 (55.3)	0.0472
McCabe score 2 or 3, *n* (%)	37 (21)	26 (17.4)	9 (60)	0.0007
Prior antimicrobial use, *n* (%)	26 (12.9)	19 (11.4)	6 (27.3)	0.0859
Prior hospitalization, *n* (%)	44 (21.5)	36 (21.3)	7 (31.8)	0.2822
Prior surgery, *n* (%)	10 (4.9)	9 (5.3)	0 (0)	0.603
Site where culture was collected				
Medical ward, *n* (%)	202 (80.5)	159 (80.3)	31 (81.6)	1
Surgical ward, *n* (%)	34 (13.5)	31 (15.7)	0 (0)	0.0065
ICU, *n* (%)	15 (6)	8 (4)	7 (18.4)	0.004
Community-acquired, *n* (%)	101 (49.5)	90 (53.3)	7 (31.8)	0.071
Presence of CVC, *n* (%)	40 (20.2)	28 (17.3)	11 (47.8)	0.002
TPN, *n* (%)	2 (1.2)	0 (0)	2 (10)	0.0149
RRT, *n* (%)	22 (10.7)	17 (10.1)	5 (21.7)	0.1522
Neutropenia, *n* (%)	7 (3.4)	4 (2.4)	3 (13)	0.0384
> or =2 SIRS, *n* (%)	171 (84.2)	138 (82.1)	21 (95.4)	0.1351
Severe sepsis, *n* (%)	40 (21.5)	25 (16.1)	10 (55.6)	0.0005
Septic shock, *n* (%)	14 (7.5)	7 (4.5)	7 (38.9)	<0.0001
Primary BSI, *n* (%)	49 (28.3)	41 (25.8)	2 (9.5)	0.0658
Infective Endocarditis, *n* (%)	14 (8.1)	13 (8.2)	0 (0)	0.2205
CLABSI, *n* (%)	26 (15)	21 (13.2)	4 (19)	0.7458
SSTI, *n* (%)	23 (13.3)	22 (13.8)	1 (4.8)	0.3142
Monomicrobial bacteremia, *n* (%)	165 (82.9)	142 (86.1)	13 (59.1)	0.0043
MRSA, *n* (%)	102 (37.9)	82 (50.6)	18 (72)	0.0336

* Data about mortality were not available for 17 patients. ** Numbers in parentheses show percentages among patients with available data. *** Continuous variables were compared using the Mann–Whitney U-test for non-normally distributed variables. All tests were two-tailed and a *p*-value equal or lower than 0.05 was considered significant. BSI: bloodstream infection; CLABSI: central-line associated bloodstream infection; CVC: central venous catheter; ICU: intensive care unit; IQR: interquartile range; MRSA: methicillin-resistant *Staphylococcus aureus*; RRT: renal replacement therapy; SIRS: systemic inflammatory response syndrome; SSTI: skin and soft tissue infection; TPN: total parenteral nutrition.

**Table 2 diagnostics-13-01975-t002:** Treatment and outcomes of patients with *Staphylococcus aureus* bacteremia in total and in regards to mortality.

Characteristic	All Patients * (*n* = 256)	Survived (*n* = 201)	Died (*n* = 38)	*p*-Value ***
Appropriate empirical treatment, *n* (%)	87 (60) **	76 (58.9)	9 (69.2)	0.5626
Duration of antimicrobial treatment, days, median (IQR)	15 (10–28)	15 (10–28)	11 (9–27)	0.2326
Definite treatment				
Vancomycin, *n* (%)	49 (37.1)	41 (36.9)	6 (40)	1
Teicoplanin, *n* (%)	9 (6.8)	5 (4.5)	4 (26.7)	0.0118
Daptomycin, *n* (%)	29 (22)	25 (22.5)	3 (20)	1
Antistaphylococcal penicillin, *n* (%)	30 (22.7)	24 (21.6)	2 (13.3)	0.7346
Duration of hospital stay, days, median (IQR)	20 (11–30)	20 (12–30)	13 (10.5–32)	0.5949
Hospital mortality, *n* (%)	38 (15.9)			

* Data about mortality were not available for 17 patients. ** Numbers in parentheses show percentages among patients with available data. *** Continuous variables were compared using the Mann–Whitney U-test for non-normally distributed variables. All tests were two-tailed and a *p*-value equal or lower than 0.05 was considered significant. IQR: interquartile range.

**Table 3 diagnostics-13-01975-t003:** Characteristics of patients with *Staphylococcus aureus* bacteremia in regards to when the bacteremia occurred.

Characteristic	Pre-COVID-19 (*n* = 194)	Post-COVID-19 (*n* = 62)	*p*-Value **
Age, years, median (IQR)	70.5 (58–82)	75 (65.8–81)	0.1515
Female gender, *n* (%)	79 (40.7) *	22 (35.5)	0.5508
McCabe score 2 or 3, *n* (%)	31 (22.1)	6 (16.7)	0.6468
Prior antimicrobial use, *n* (%)	16 (11.2)	10 (17.2)	0.2534
Prior hospitalization, *n* (%)	30 (20.7)	14 (23.3)	0.71
Prior surgery, *n* (%)	4 (2.8)	6 (9.8)	0.0677
Site where culture was collected			
Medical ward, *n* (%)	154 (80.6)	48 (80)	0.7239
Surgical ward, *n* (%)	24 (12.6)	10 (16.7)	0.5189
ICU, *n* (%)	13 (6.8)	2 (3.3)	0.5338
Community-acquired, *n* (%)	77 (53.1)	24 (40.7)	0.1236
Presence of CVC, *n* (%)	25 (17.4)	15 (27.8)	0.1147
TPN, *n* (%)	2 (1.6)	0 (0)	1
RRT, *n* (%)	15 (10.5)	7 (11.3)	1
Neutropenia, *n* (%)	7 (4.9)	0 (0)	0.1043
> or =2 SIRS, *n* (%)	117 (83)	54 (87.1)	0.5348
Severe sepsis, *n* (%)	27 (21.6)	13 (21.3)	1
Septic shock, *n* (%)	6 (4.8)	8 (13.3)	0.0697
Primary BSI, *n* (%)	31 (27)	18 (31)	0.5952
Infective Endocarditis, *n* (%)	5 (4.3)	9 (15.5)	0.0168
CLABSI, *n* (%)	24 (20.9)	2 (3.4)	0.0016
SSTI, *n* (%)	13 (11.3)	10 (17.2)	0.3432
Monomicrobial bacteremia, *n* (%)	119 (84.4)	46 (79.3)	0.4106
MRSA, *n* (%)	70 (46.7)	25 (51)	0.6242

* Numbers in parentheses show percentages among patients with available data. ** Continuous variables were compared using the Mann–Whitney U-test for non-normally distributed variables. All tests were two-tailed and a *p*-value equal to or lower than 0.05 was considered significant. BSI: bloodstream infection; CLABSI: central-line associated bloodstream infection; CVC: central venous catheter; ICU: intensive care unit; IQR: interquartile range; MRSA: methicillin-resistant *Staphylococcus aureus*; RRT: renal replacement therapy; SIRS: systemic inflammatory response syndrome; SSTI: skin and soft tissue infection; TPN: total parenteral nutrition.

**Table 4 diagnostics-13-01975-t004:** Treatment and outcomes of patients with *Staphylococcus aureus* bacteremia in regards to when the bacteremia occurred.

Characteristic	Pre-COVID-19 (*n* = 194)	Post-COVID-19 (*n* = 62)	*p*-Value **
Appropriate empirical treatment, *n* (%)	68 (62.4) *	19 (52.8)	0.3315
Duration of antimicrobial treatment, days, median (IQR)	14 (9–25.3)	28 (14–31)	0.0256
Definite treatment			
Vancomycin, *n* (%)	38 (42.2)	11 (26.2)	0.0848
Teicoplanin, *n* (%)	9 (10)	0 (0)	<0.0001
Daptomycin, *n* (%)	16 (17.8)	13 (31)	0.1143
Antistaphylococcal penicillin, *n* (%)	19 (21.1)	11 (26.2)	0.5126
Duration of hospital stay, days, median (IQR)	16 (10–25)	27 (15–44)	0.0009
Hospital mortality ***, *n* (%)	29 (16.1)	9 (15.3)	1

* Numbers in parentheses show percentages among patients with available data. ** Continuous variables were compared using the Mann–Whitney U-test for non-normally distributed variables. All tests were two-tailed and a *p*-value equal or lower than 0.05 was considered significant. *** Data about mortality were not available for 17 patients. IQR: interquartile range.

**Table 5 diagnostics-13-01975-t005:** Results of the regression analysis regarding patient mortality at 30 days after the occurrence of *Staphylococcus aureus* bacteremia.

Characteristic	Univariate Analysis *p*	Multivariate Analysis *p*	OR (95% CI)
Female gender	0.0386	0.095	5.144 (0.751–35.210)
Age (per year)	0.001	0.087	1.101 (0.986–1.230)
McCabe score	0.0002	0.067	3.129 (0.924–10.594)
Prior antimicrobial use	0.0401	0.375	3.516 (0.218–56.578)
CVC	0.0009	0.077	10.333 (0.775–137.77)
Neutropenia	0.0102	0.280	8.918 (0.169–471.592)
Severe sepsis	<0.0001	0.050	12.294 (1.005–150.354)
Septic shock	<0.0001	0.007	57.180 (3.051–1071.664)
Monomicrobial BSI	0.0055	0.353	3.001 (0.295–30.483)
MRSA	0.03	0.255	3.045 (0.448–20.677)
Community-acquired	0.059	0.447	2.157 (0.297–15.669)

BSI: bloodstream infection; CI: confidence intervals; CVC: central venous catheter; MRSA: methicillin-resistant *Staphylococcus aureus*; OR: odds ratio.

## Data Availability

The data presented in this study are available on request from the corresponding authors.

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
