# Peer review of "A Retrospective Study of Staphylococcus aureus Bacteremia in a Tertiary Hospital and Factors Associated with Mortality"

_diagnostics, 2023, doi:10.3390/diagnostics13111975_

Round 1
Reviewer 1 Report
Dear authors
This manuscript has evaluated SAB risk factors in Greece during 2005-2021.
There are following comments for this manuscript:
1- The title and abstract of the manuscript are appropriate and relative to the text.
2- The introduction section can be improved in the aspect of risk factors associated with S. aureus bacteremia and mortality rates.
3- Methods is suitable. However, if you have received ethical code, please add in this section. Please correct “36oC” to 36°C. Additionally, 36°C is lower than optimal temperature for growth of bacterial pathogens.
4- The time duration of the study from the year 2005 to 2021 gives valuable data, though it seems that some techniques have been changed, possibly even in this study, for bacterial identification and disease diagnosis from 2005.
5- Please add p-value scores in addition to percentages in parentheses.
6- Please define “McCabe score” and also abbreviations in the text when used for the first time.
7- The writing of the manuscript is appropriate and needs minor revision.
8- Results are appropriate; however, follow journal style for tables and definition of abbreviations placement related to tables.
9- If MRSA has been also detected, please add the results.
10- Please use more recent studies from 2021 in the discussion section regarding MRSA and MSSA bacteremia, risk factors and management/control strategies. Please also consider the following publication:
· Nickel Nanoparticles: Applications and Antimicrobial Role against Methicillin-Resistant Staphylococcus aureus Infections. Antibiotics. 2022 Sep 7;11(9):1208
·
With best regards
Dear Dr
Thank you very much
This manuscript has evaluated SAB risk factors in Greece during 2005-2021.
There are following comments for this manuscript:
1- The title and abstract of the manuscript are appropriate and relative to the text.
2- The introduction section can be improved in the aspect of risk factors associated with S. aureus bacteremia and mortality rates.
3- Methods is suitable. However, if you have received ethical code, please add in this section. Please correct “36oC” to 36°C. Additionally, 36°C is lower than optimal temperature for growth of bacterial pathogens.
4- The time duration of the study from the year 2005 to 2021 gives valuable data, though it seems that some techniques have been changed, possibly even in this study, for bacterial identification and disease diagnosis from 2005.
5- Please add p-value scores in addition to percentages in parentheses.
6- Please define “McCabe score” and also abbreviations in the text when used for the first time.
7- The writing of the manuscript is appropriate and needs minor revision.
8- Results are appropriate; however, follow journal style for tables and definition of abbreviations placement related to tables.
9- If MRSA has been also detected, please add the results.
10- Please use more recent studies from 2021 in the discussion section regarding MRSA and MSSA bacteremia, risk factors and management/control strategies. Please also consider the following publication:
· Nickel Nanoparticles: Applications and Antimicrobial Role against Methicillin-Resistant Staphylococcus aureus Infections. Antibiotics. 2022 Sep 7;11(9):1208
·
With best regards
Author Response
Reviewer 1
Dear authors
This manuscript has evaluated SAB risk factors in Greece during 2005-2021.
There are following comments for this manuscript:
1- The title and abstract of the manuscript are appropriate and relative to the text.
Response: Thanks for the comment.
2- The introduction section can be improved in the aspect of risk factors associated with S. aureus bacteremia and mortality rates.
Response: Thanks for the comment. We expanded the introduction section to increase the information provided regarding the aspects mentioned by the reviewer as can be seen in the revised version of the manuscript.
3- Methods is suitable. However, if you have received ethical code, please add in this section. Please correct “36oC” to 36°C. Additionally, 36°C is lower than optimal temperature for growth of bacterial pathogens.
Response: Thanks. We added that in the methods section and made that correction. As for the temperature, we agree with the reviewer. Most of the bacterial pathogens are mesophilic in nature. Their optimal growth temperatures range from room temperature (about 20 °C) to about 45 °C. The most ideal for bacterial growth, including normal human microbiota and human pathogens, is the human body temperature of 36-37 °C.
4- The time duration of the study from the year 2005 to 2021 gives valuable data, though it seems that some techniques have been changed, possibly even in this study, for bacterial identification and disease diagnosis from 2005.
Response: Thanks for the comment. We added this as a limitation at the end of the discussion section.
5- Please add p-value scores in addition to percentages in parentheses.
Response: Thanks for the comment. However, we do not exactly understand what the reviewer means with this comment. All comparisons of different parameters in the present manuscript are shown in tables where in the last column, a p-value of the statistical comparison is presented. Thus, we do not understand what modification is required with this comment. If an explanation is provided regarding what is needed, we could consider that in a future revision.
6- Please define “McCabe score” and also abbreviations in the text when used for the first time.
Response: Thanks for the comment. We defined McCabe score in the methods section, where a reference had already been placed to allow the reader to look up to a study that already used that before. We also made the requested modification in some terms where abbreviations were used without explanation in the previous version of this manuscript.
7- The writing of the manuscript is appropriate and needs minor revision.
Response: Thanks for the comment. We performed minor revision of the English grammar and syntax, as suggested.
8- Results are appropriate; however, follow journal style for tables and definition of abbreviations placement related to tables.
Response: Thanks for the comment. We looked up again the instructions for authors and made some corrections in the abbreviations of the tables as can be seen in the revised version of the manuscript.
9- If MRSA has been also detected, please add the results.
Response: Thanks for the comment. Data on MRSA are already in the results section. More specifically, the data are shown at the end of Table 1 and Table 3, where a statistical comparison among different groups is also shown. Comments on that are also shown at the end of the first and the second paragraph of the results section.
10- Please use more recent studies from 2021 in the discussion section regarding MRSA and MSSA bacteremia, risk factors and management/control strategies. Please also consider the following publication:
- Nickel Nanoparticles: Applications and Antimicrobial Role against Methicillin-Resistant Staphylococcus aureus Infections. Antibiotics. 2022 Sep 7;11(9):1208
Response: Thanks for the comment. We made some modifications in the discussion section, added a new paragraph about treatment and added the abovementioned reference. We also added some more recent references, as can be seen in the revised version of the manuscript.
Reviewer 2 Report
In this study, they evaluated a cohort of patients with SAB hospitalized in a tertiary hospital and identified factors independently associated with mortality. The study is limited to one center, and has several observations that need to be corrected.
While the findings in this study are interesting, similar studies have been performed elsewhere and the findings in this study represent only a modest incremental advance over the data in the literature.
Also, I suggest reviewing the scope of the "Diagnostics" Journal https://www.mdpi.com/journal/diagnostics/about. In this study, clinical factors are evaluated, the prognosis or diagnosis is not studied.
Here are my major comments.
Methods
Include the STROBE checklist and verify if they comply with all the items.
What were the inclusion and exclusion criteria? only patients with positive culture for S. aureus? or positive culture and bacteremia?
What was the definition of bacteremia?
The authors do not explain the primary and secondary outcomes in the retrospective cohort. I suggest checking out STROBE.
Statistic analysis.
What was the confidence interval used and the p-value to consider significant.
Why did you use p lower than or equal to 0.1?
Confounding variables for multivariate analysis are not explained.
Results
Include the STROBE flowchart for the enrollment of participants.
There were no exclusions of the cases?
Table 1. At the bottom of the table it is not explained what statistical tests were performed.
Table 2. They have made comparisons with Chi square or T-test, I don't think they are tests to compare treatment and results. Authors should make comparisons with HR or RR, keep in mind that it is treatment and outcomes.
Tables 1, 3 and 4. For the base characteristics it is fine, but they are not the most appropriate p-value analyses. Comparisons should be made with OR, RR, or HR, with crude and adjusted analyses. Based on those findings. Significant results with a p-value less than 0.05 should perform adjusted analyses.
Table 5. Why are the results of p=0.05 shown to be significant? p-value should be less than 0.05. In the Table it is not understood if the OR is raw or adjusted. If they are adjusted analysis, what were the adjustment variables.
Where are the results of the sensitivity analysis for the fitted multivariate models shown?
Why were comorbidities not included in the analyses?
Were there cases associated with COVID-19 infection? if there were these cases they were included or excluded.
Discussion
The study has several limitations that must be discussed.
They should include a paragraph with the implications of their statements for clinical practice.
The conclusions should be improved on the basis of the previous comments.
Minor comments
The title should include the study design (see STROBE).
Author Response
In this study, they evaluated a cohort of patients with SAB hospitalized in a tertiary hospital and identified factors independently associated with mortality. The study is limited to one center, and has several observations that need to be corrected.
While the findings in this study are interesting, similar studies have been performed elsewhere and the findings in this study represent only a modest incremental advance over the data in the literature.
Response: Thanks for the comment. Indeed, this is true, however, several differences are noted in different geographic areas in the case of infectious diseases and clinical microbiology, and this makes every study valuable even if no major novel findings are noted. Thus, we feel that this manuscript merits publication, since it provides data for a prolonged period of study regarding a serious infection.
Also, I suggest reviewing the scope of the "Diagnostics" Journal https://www.mdpi.com/journal/diagnostics/about. In this study, clinical factors are evaluated, the prognosis or diagnosis is not studied.
Response: This is an important comment, and we thank the reviewer for bringing it into discussion. Indeed, when reading the scope of the journal ‘Diagnostics’, at first look, our manuscript may sound as being out of scope. However, this manuscript was submitted specifically to the special issue ‘Antimicrobial Resistance during and after COVID-19’ of the ‘Diagnostics’ journal (https://www.mdpi.com/journal/diagnostics/special_issues/3TZG77F099). The guest editors in their call for papers end with the following sentence: ‘With this Special Issue, we would like to explore the impact of the COVID-19 pandemic on antibacterial resistance, antimicrobial use, nosocomial infections and infection control and discuss methods and strategies to improve policies and practices in the aftermath of the pandemic’. Thus, our manuscript is within the scope of the special issue, as it provides the above-mentioned information about S. aureus bacteremia in general and also in regards to whether the infection occurred before or after the onset of the COVID-19 pandemic.
Here are my major comments.
Methods
Include the STROBE checklist and verify if they comply with all the items.
Response: Thanks for the comment. We have revised the manuscript based on the STROBE checklist and added in the supplementary material a comprehensive table (Table S1) mentioning some necessary information regarding methods and the results as per the STROBE checklist.
What were the inclusion and exclusion criteria? only patients with positive culture for S. aureus? or positive culture and bacteremia?
Response: The only inclusion criterion was the presence of bacteremia by S. aureus, as was noted at the beginning of the methods section in the first version of the manuscript. To make it clearer for the reader to understand, we explained it more at that part and also added a definition of bacteremia. There was no exclusion criterion in the present study.
What was the definition of bacteremia?
Response: We defined bacteremia to make it clearer for the reader to understand in the revised version of the manuscript (bacteremia by S. aureus was defined as the presence of at least one positive blood culture for S. aureus).
The authors do not explain the primary and secondary outcomes in the retrospective cohort. I suggest checking out STROBE.
Response: Thanks. We checked the STROBE statement. We made several changes in the manuscript, as can be seen in the revised version. Among those changes was also the addition of clear primary and secondary outcomes of this retrospective cohort study, as can be seen in the methods section of the present manuscript.
Statistic analysis.
What was the confidence interval used and the p-value to consider significant.
Response: Thanks. We added that information regarding the multivariate regression analysis in the statistics subsection of the methods section, as can be seen in the revised version of the manuscript.
Why did you use p lower than or equal to 0.1?
Response: Thanks for the comment. Traditionally, a significance level of p < 0.05 is commonly used in many fields as the threshold for statistical significance. However, in some cases, using a significance level of p < 0.05 may result in too many false negatives (i.e., type II errors) or too few false positives (i.e., type I errors). In such cases, some researchers use a higher significance level, such as p < 0.10, to increase the power of the analysis and decrease the risk of missing important variables to be used later on in a multivariate logistic regression analysis. This is what we did in the present study.
Confounding variables for multivariate analysis are not explained.
Response: Thanks. We have added a comment in the last paragraph of the discussion section, where we mention the limitations of the present study, and we specifically mention the factors that could be confounding variables for the multivariate analysis. These factors could be age, presence of sepsis, severity of underlying disease, and antimicrobial resistance. This can be seen at the end of the discussion section of the revised manuscript.
Results
Include the STROBE flowchart for the enrollment of participants.
Response: This is not applicable in the present study. Patients who had at least one positive blood culture were enrolled. The patients’ identifying information (names, patients’ codes) were retrieved from the microbiology department and the rest of data were retrieved from the hard copies of the patients’ notes and the hospital’s electronic system. No patients were excluded.
There were no exclusions of the cases?
Response: No. There were no patients excluded in the present analysis. We added that information at the beginning of the results section, in the first paragraph in the revised version of the manuscript.
Table 1. At the bottom of the table it is not explained what statistical tests were performed.
Response: Even though we do not agree with this comment, we complied with the reviewer’s suggestion. We have added the statistical analysis at the footnote of all tables except from the one with the regression analysis.
Table 2. They have made comparisons with Chi square or T-test, I don't think they are tests to compare treatment and results. Authors should make comparisons with HR or RR, keep in mind that it is treatment and outcomes.
Response: Thanks for the comment and for giving the opportunity to clarify this. First of all, since the sample size is small, we didn’t use the Chi square test. For small numbers, Fisher’s exact test is indicated. Fisher's exact test does not directly provide estimates of odds ratios (OR) or hazard ratios (HR). Instead, it examines the independence between two variables by testing the null hypothesis that there is no association between them. If someone is interested in estimating the strength of the association between treatment and mortality (for example), he should typically use other statistical methods such as logistic regression for OR or survival analysis (e.g., Cox proportional hazards model) for HR. These methods can provide him with estimates of the effect size, confidence intervals, and p-values associated with the treatment-mortality relationship. In summary, while Fisher's exact test can determine whether there is a statistically significant association between two parameters, it does not provide estimates of OR or HR. Appropriate regression models are indicated.
Tables 1, 3 and 4. For the base characteristics it is fine, but they are not the most appropriate p-value analyses. Comparisons should be made with OR, RR, or HR, with crude and adjusted analyses. Based on those findings. Significant results with a p-value less than 0.05 should perform adjusted analyses.
Response: We are skeptical about this. First of all, as per the previous comment, OR, RR and HR cannot be reliably estimated with the Fisher’s exact test. On the other hand, we, as a research group, have several examples where an association was not identified in this type of statistics, but was identified with a univariate linear regression analysis, as performed in the present study (and in other studies performed by our group). This type of analysis performed in the present study is explained in detail in the statistics subsection in the methods section, and in the results section in the subsection ‘Regression analysis of in-hospital mortality among patients with SAB’. Even though these two analyses usually show similar results, we feel that the type of analysis performed in our study has a relatively higher sensitivity to identify factors associated with the outcome chosen. Moreover, we feel that adding more parameters, such as ORs, RRs or HRs would only complicate things and make the manuscript harder for the reader to understand. This is why we did not calculate the ratios requested by the reviewer.
Table 5. Why are the results of p=0.05 shown to be significant? p-value should be less than 0.05. In the Table it is not understood if the OR is raw or adjusted. If they are adjusted analysis, what were the adjustment variables.
Response: Thanks for the comment. We understand it may sound weird. However, the 95% confidence intervals for the ORs are all above 1, thus, we present it as statistically significant. We used logistic regression function in SPSS, thus, the odds ratio (exp(B)) are adjusted for all the other variables in the analysis, which are all the parameters shown in Table 5.
Where are the results of the sensitivity analysis for the fitted multivariate models shown?
Response: Thanks for the comment. We added and removed predictor variables, thus, creating alternative models to observe how the coefficients and significance levels change and there were no great changes in the alternative models. However, we feel that this comment by the reviewer is very technical and unnecessary to show, since it will definitely create confusion to the reader who might not be that experienced in statistical analysis.
Why were comorbidities not included in the analyses?
Response: Thanks for the comment. Since there were missing values for the comorbidities in quite some patients, we felt that adding them in the analysis may have been misleading. Thus, we did not add them.
Were there cases associated with COVID-19 infection? if there were these cases they were included or excluded.
Response: Yes, there were such cases. There were no exclusion criteria, thus, these cases were not excluded and are presented in the present cohort.
Discussion
The study has several limitations that must be discussed.
Response: Indeed, this study has some notable limitations. We had added such a paragraph at the end of the discussion section, and now, we have expanded that paragraph to include some limitations that were noted by the reviewers. This paragraph can be seen at the end of the discussion section, just before the conclusions section in the revised version of the manuscript.
They should include a paragraph with the implications of their statements for clinical practice.
Response: Thanks. We added a paragraph with implications for clinical practice, as suggested by the reviewer at the end of the discussion section, just above the limitations. We feel that it is now clear for the reader that clinicians caring for patients with severe sepsis or septic shock in general should consider the possibility of SAB, while, for patients with SAB and such a severe clinical presentation, optimization of medical care along with source control are required to reduce mortality.
The conclusions should be improved on the basis of the previous comments.
Response: Thanks for the comment. We have modified the conclusions section of the manuscript based on the abovementioned comments as can be seen in the revised version of the manuscript.
Minor comments
The title should include the study design (see STROBE).
Response: Thanks. We modified the title, as suggested by the STROBE statement as can be seen in the revised version of the manuscript.
Reviewer 3 Report
The study by Ioannou and colleagues report factors independently associated with mortality of inpatients diagnosed with Staphylococcus aureus bacteremia. The authors retrospectively analyzed 256 patients from a Greek hospital during 2005-2021. In general, the manuscript is well written and tecnically sound.
Few questions need to be clarified:
-What do the authors consider "appropriate empirical treatment"?
Although the authors comment on the use of some agents as drugs of choice for the therapy, the antimicrobial susceptibility profile was not shown. These data can help in understanding the appropriate treatment. I recommend presenting the antimicrobial susceptibility profile of the isolates.
-What do the values in parentheses in tables 2 and 4 mean? The percentage was calculated against which n total value?
-Authors should describe the meaning of acronyms in the text in the first citation. For example, the meaning of CVC, TPN and others are shown in Table 1.
Author Response
The study by Ioannou and colleagues report factors independently associated with mortality of inpatients diagnosed with Staphylococcus aureus bacteremia. The authors retrospectively analyzed 256 patients from a Greek hospital during 2005-2021. In general, the manuscript is well written and tecnically sound.
Response: Thanks for the comment. We hope the reviewer finds the revised version of the manuscript more concise and more informative.
Few questions need to be clarified:
-What do the authors consider "appropriate empirical treatment"?
Response: Thanks for the comment. This term was defined in the methods section, at the end of 2.1 subsection, just before 2.2 subsection. More specifically, ‘An empirical treatment was deemed as appropriate if it included at least one antimicrobial agent active against the S. aureus strain that was eventually identified.’
Although the authors comment on the use of some agents as drugs of choice for the therapy, the antimicrobial susceptibility profile was not shown. These data can help in understanding the appropriate treatment. I recommend presenting the antimicrobial susceptibility profile of the isolates.
Response: Thanks for the comment. We have read carefully all the reviewer’s comments, including this, and we do agree that presenting the exact antimicrobial susceptibility data would have been ideal. However, due to the retrospective nature of the present study, the exact MICs are not available. The basic focus of the present study’s researchers was to collect data regarding methicillin resistance, as this is the most defining step in the management of S. aureus bacteremia. Notably, however, no VRSA strain had ever been identified in our microbiology in the University Hospital of Heraklion.
-What do the values in parentheses in tables 2 and 4 mean? The percentage was calculated against which n total value?
Response: Thanks for the comment. We added a sentence in the footnotes. The values in parentheses show percentages among patients with available data for that particular characteristic shown in the left column.
-Authors should describe the meaning of acronyms in the text in the first citation. For example, the meaning of CVC, TPN and others are shown in Table 1.
Response: Thanks for the comment. CVC was explained earlier, in the methods section. We added some detailed explanation for some abbreviations that had not been explained in the first version of the manuscript.
Reviewer 4 Report
Staphylococcus aureus bacteremia in a tertiary hospital and 2 factors associated with mortality by Petros Ioannou at all is a very important clinical set of data and well-written MS. It deserves to be popularised, illuminating the real problem in the clinical practice of treating SA bacteremia.
Labe 61-66
“Before the antibiotic era, this infection had a 61 mortality that could be close to 80% [13]. With the development of current antimicrobial 62 treatment mortality rates are becoming quite lower, reaching 20% for 30-day mortality, 63 while according to recent studies, infection-related mortality reaches 13% [14]. A recent 64 systematic review showed that SAB mortality may have decreased in the last decades, 65 however, more than 25% of patients will die within three months, thus, there is still a need 66 for further improvement [15].”
That is an unfortunate reality but with frequent repetition of the same issue: the technical problem of timely detection of resistance and hetero-resistance (see some seminal papers below), even when repetitive blood cultures are taken. The origin of the wrong laboratory practice is in the clonal/sub-culturing step, where hetero-resistance is lost…
See reference below
https://www.sciencedirect.com/science/article/pii/S1198743X14633322
https://journals.asm.org/doi/10.1128/AAC.38.4.724
and the explanation of what needs to be changed (now) in order to get a more realistic picture from blood culture data.
https://pubmed.ncbi.nlm.nih.gov/32973694/
In Lane, 221-222 is written: “Paradoxically, in the present study, the 221 rate of appropriate empirical treatment was not significantly different among patients 222 who survived and those who died.” This statement if followed by (lane 262) “present observations has to do with the finding that empirical treatment was not 262 adequate in about 40% of patients.”
That is a moment for authors to add a brave claim: there is a clear inherent limitation of treatment strategy based only on empirical data. There is a clear need for evidence-based laboratory data (like blood antibiotic MIC values…) from multiple time points during hospitalization.
I would like the authors to include the problem of unresolved hetero-resistance and enhance their conclusion more often in the text (abstract, introduction discussion and conclusion): multiple blood cultures must be performed during bacteremia, in order to follow the evolution of antibiotic resistance in time…
Minor comment:
Lane 99
360C should use superscript
Author Response
Staphylococcus aureus bacteremia in a tertiary hospital and 2 factors associated with mortality by Petros Ioannou at all is a very important clinical set of data and well-written MS. It deserves to be popularised, illuminating the real problem in the clinical practice of treating SA bacteremia.
Response: Thanks for the nice comments. We hope the reviewer finds the revised version of the manuscript more concise and informative.
Labe 61-66
“Before the antibiotic era, this infection had a 61 mortality that could be close to 80% [13]. With the development of current antimicrobial 62 treatment mortality rates are becoming quite lower, reaching 20% for 30-day mortality, 63 while according to recent studies, infection-related mortality reaches 13% [14]. A recent 64 systematic review showed that SAB mortality may have decreased in the last decades, 65 however, more than 25% of patients will die within three months, thus, there is still a need 66 for further improvement [15].”
That is an unfortunate reality but with frequent repetition of the same issue: the technical problem of timely detection of resistance and hetero-resistance (see some seminal papers below), even when repetitive blood cultures are taken. The origin of the wrong laboratory practice is in the clonal/sub-culturing step, where hetero-resistance is lost…
See reference below
https://www.sciencedirect.com/science/article/pii/S1198743X14633322
https://journals.asm.org/doi/10.1128/AAC.38.4.724
and the explanation of what needs to be changed (now) in order to get a more realistic picture from blood culture data.
https://pubmed.ncbi.nlm.nih.gov/32973694/
Response: Thanks for the comment. We studied very carefully the references provided and we added a paragraph regarding the issue of heteroresistance and how it may affect the treatment of patients with SAB. This can be seen in line 326 in the discussion section of the revised manuscript.
In Lane, 221-222 is written: “Paradoxically, in the present study, the 221 rate of appropriate empirical treatment was not significantly different among patients 222 who survived and those who died.” This statement if followed by (lane 262) “present observations has to do with the finding that empirical treatment was not 262 adequate in about 40% of patients.”
That is a moment for authors to add a brave claim: there is a clear inherent limitation of treatment strategy based only on empirical data. There is a clear need for evidence-based laboratory data (like blood antibiotic MIC values…) from multiple time points during hospitalization.
Response: Thanks for the comment. We expanded the discussion section to include the abovementioned point so that the reader can understand that, indeed, empirical data alone are inadequate. This can be seen in the discussion section of the manuscript, starting in line 321.
I would like the authors to include the problem of unresolved hetero-resistance and enhance their conclusion more often in the text (abstract, introduction discussion and conclusion): multiple blood cultures must be performed during bacteremia, in order to follow the evolution of antibiotic resistance in time…
Response: Thanks. We have changed some parts in the abstract, the introduction, the discussion, and the conclusions section to comment on the issue of heteroresistance, as suggested by the reviewer. This can be seen in the corresponding parts of the revised version of the manuscript. The addition in the discussion section is the most detailed one.
Minor comment:
Lane 99
360C should use superscript
Response: Thanks. We corrected that.
Round 2
Reviewer 2 Report
No comments